# Associations between long-term drought and diarrhea among children under five in low- and middle-income countries

Pin Wang [1,2 ✉], Ernest Asare [3], Virginia E. Pitzer [3], Robert Dubrow [1,2,4] & Kai Chen [1,2,4]

Climate change is projected to intensify drought conditions, which may increase the risk of diarrheal diseases in children. We constructed log-binomial generalized linear mixed models to examine the association between diarrhea risk, ascertained from global-scale nationally representative Demographic and Health Surveys, and drought, represented by the standardized precipitation evapotranspiration index, among children under five in 51 low- and middle-income countries (LMICs). Exposure to 6-month mild or severe drought was associated with an increased diarrhea risk of 5% (95% confidence interval 3–7%) or 8% (5–11%), respectively. The association was stronger among children living in a household that needed longer time to collect water or had no access to water or soap/detergent for handwashing. The association for 24-month drought was strong in dry zones but weak or null in tropical or temperate zones, whereas that for 6-month drought was only observed in tropical or temperate zones. In this work we quantify the associations between exposure to long-term drought and elevated diarrhea risk among children under five in LMICs and suggest that the risk could be reduced through improved water, sanitation, and hygiene practices, made more urgent by the likely increase in drought due to climate change.

[1] Department of Environmental Health Sciences, Yale School of Public Health, New Haven, CT, USA. [2] Yale Center on Climate Change and Health, Yale School of Public Health, New Haven, CT, USA. [3] Department of Epidemiology of Microbial Diseases and the Public Health Modeling Unit, Yale School of Public Health, New Haven, CT, USA. [4] These authors jointly supervised this work: Robert Dubrow, Kai Chen. ✉email: pin.wang@yale.edu

Climate change impacts humans from infancy and adolescence to adulthood and old age[1], and it is projected to profoundly intensify hydrological variability and increase the intensity and frequency of extreme weather events, including floods and droughts[2], amplifying the public health threat of food- and water-borne enteric diseases[3,4]. Diarrheal diseases are an unequivocal threat to children's health. A systematic analysis in 195 countries for the Global Burden of Disease Study in 2016 estimated that diarrhea was the fifth leading cause of death among children younger than five years, causing 446 thousand deaths globally[5]. Of these, the greatest burden (90%) occurred in areas with poor water, sanitation, and hygiene (WASH) practices such as sub-Saharan Africa and South Asia[6]. In addition to causing mortality, diarrhea in children can also have lasting adverse effects such as impaired growth and cognitive development and increased susceptibility to chronic diseases[7]. The World Health Organization estimated that in 2050, climate change could be responsible for approximately 32,954 additional diarrheal deaths worldwide among children aged 0–15 years[8]. Understanding relationships between climate and diarrheal disease is critical because even a small increase in the risk of diarrhea represents a considerable disease burden[3], especially in low- and middle-income countries (LMICs).

In general, positive associations have been observed between ambient temperature and all-cause or bacterial diarrhea, whereas negative associations have been observed between ambient temperature and viral diarrhea, particularly rotavirus-associated gastroenteritis[3,9]. Since drought could elevate the risk of diarrhea by concentrating pathogens in water sources, it is reasonable to expect that a lower volume of rainfall is likely to be associated with a higher risk of diarrhea. In addition, extreme precipitation events and accompanying floods that spread sewage and animal waste are known to be associated with an increased risk of diarrhea[10]. Thus, the relationship between rainfall and diarrhea risk is complex, and it is not surprising that studies of this relationship have had conflicting results, with various studies finding a higher risk of diarrhea to be associated with both low and high levels of rainfall[11–13], moderate rainfall[14,15], or only the highest[16] or lowest[17,18] rainfall volumes.

Drought is a more complex phenomenon than simply low levels of rainfall and high temperatures, and researchers have struggled to agree upon its definitions[19]. Although the World Meteorological Organization provided a simple definition of drought as "prolonged absence or marked deficiency of precipitation" in 1992[20], recent advances have been made regarding the importance of integrating water demand into the definition, in addition to taking water supply into account. Demand, which is determined by evapotranspiration, is more difficult to measure than supply[21]. Apart from the lack of consensus on the definition, particularly sparse epidemiological evidence[3] further hinders our understanding of the relationship between more complex measures of drought and diarrheal diseases. In place of rainfall, aridity, calculated as the ratio of daily minimum temperature over specific humidity, was employed in a recent spatiotemporal analysis in Afghanistan, reporting a positive association between aridity and diarrhea incidence[22]. To date, to our knowledge, this is the only study to assess the relationship between a more complex measure of drought and diarrhea risk.

As expected, effective WASH practices have been found to be associated with a lower risk of diarrheal diseases[23,24]. The changing climate, especially increased drought, might threaten access to and uptake of effective WASH, resulting in increased diarrhea risk[25]. Additionally, the shortage of clean water during drought may result in a reduction in water availability for personal hygiene and sanitation, as adequate water for drinking needs to be prioritized, resulting in increased exposure to enteric pathogens.

Therefore, WASH practices could play both an interacting and mediating role in the relationship between drought and diarrhea incidence.

Notwithstanding some evidence on the association of low rainfall with increased diarrhea incidence, current knowledge on the association of more advanced measures of drought with diarrheal diseases is scarce. In this study, we aimed (1) to quantify the relationship between drought occurrence as measured by the recently-developed standardized precipitation evapotranspiration index (SPEI)[21,26] and the risk of diarrhea among children under age 5 in LMICs, and (2) to investigate whether WASH practices or other variables mediate or modify this association.

## Results

**Descriptive summary.** Among 141 surveys conducted across 51 countries, 102 surveys were located in sub-Saharan Africa (34 countries), 24 were in South and Southeast Asia (nine countries), and 15 were in Latin America and the Caribbean (eight countries). In addition, 21, 51, and 69 surveys were conducted during 1990–1999, 2000–2009, and 2010–2019, respectively. There was a total of 1,379,566 children under age five surveyed. The percentages of children having diarrhea in the previous two weeks based on different baseline characteristics, drought severity, and WASH practices are shown in Table 1, and the diarrhea incidence rate in the latest survey for each country is presented in Fig. 1. The overall incidence among children under age five included in all 141 surveys in 51 countries was 14.4% and was highest among children aged 6–23 months. Niger had the highest incidence rate (36.4%) in the most recent survey among all 51 countries, followed by Bolivia (25.1%), Liberia (23.8%), Central African Republic (22.7%), Burundi (21.4%), Malawi (20.7%), and Haiti (20.2%) (Fig. 1). The numbers of children under five who were exposed to mild drought at all timescales were about three times that of those who were exposed to severe drought during the 30-year period (Table 1). Because few children lived in continental or polar climate zones, we only fit interaction models with the tropical, dry, and temperate categories to ensure model stability.

Figure 2 shows the drought-month distribution at different timescales across all 10-km-grids in all countries, with their corresponding spatial distributions displayed in Supplementary Figs. 1–4. The 24-month drought showed a smaller median and a larger variance than droughts at the other three timescales. Furthermore, across all studied regions of the world, fewer 24-month droughts were observed than 6-, 12-, or 18-month droughts, particularly in sub-Saharan Africa. We calculated the number of drought-months each included child experienced and found the number of months with severe drought to be substantially lower than the number with mild drought (Supplementary Table 1). However, we did not observe noticeable differences in drought-months experienced by children across timescales or climate zones.

**Risk of diarrhea associated with drought events.** Both the crude (only adjusted for meteorological parameters, seasonality, and long-term trend) and main (further adjusted for baseline characteristics) models for each drought timescale were fit using the same data sample ($N = 713,918$, without missing values), comprising 80 surveys in 43 countries. The baseline characteristics of this sample are shown in Supplementary Table 2. The associations between the risk of diarrhea and drought at different timescales are presented in Fig. 3. For both the crude and main models, the associations with mild drought were significant and largely consistent across timescales, while the strongest and only significant association with severe drought was observed at the

**Table 1 Percentage of children under age five who had diarrhea in the two weeks preceding the interview in 51 countries during 1990–2019.**

| | Diarrhea incidence (%) | Number of children | Missing (%) | | Diarrhea incidence (%) | Number of children | Missing (%) |
|---|---|---|---|---|---|---|---|
| Total | 14.4 | 1,379,566 | | | | | |
| Age | | | 23.5 | Round-trip time to collect water | | | 11.5 |
| <6 months | 11.3 | 113,508 | | On premises | 12.3 | 425,111 | |
| 6–11 months | 23.3 | 115,984 | | <30 min | 15.2 | 627,539 | |
| 12–23 months | 22.3 | 219,049 | | ≥30 min | 17.2 | 168,677 | |
| 24–35 months | 14.8 | 209,305 | | Type of toilet facility | | | 3.4 |
| 36–47 months | 9.3 | 202,998 | | Improved | 12.6 | 439,452 | |
| 48–59 months | 6.8 | 194,034 | | Unimproved | 15.2 | 893,149 | |
| Sex | | | 0 | Place to wash hands | | | 41.1 |
| Male | 14.9 | 700,568 | | Fixed | 11.4 | 440,910 | |
| Female | 13.9 | 678,998 | | Mobile | 14.5 | 250,828 | |
| Residence | | | 0 | Water at the handwashing site | | | 56.8 |
| Urban | 13.4 | 408,245 | | No | 15.1 | 176,933 | |
| Rural | 14.8 | 971,321 | | Yes | 11.5 | 418,655 | |
| Wealth quintile | | | 29.9 | Soap/detergent at handwashing site | | | 57.4 |
| Lowest | 15.5 | 193,311 | | No | 13.8 | 307,703 | |
| Second | 14.6 | 193,467 | | Yes | 11.5 | 280,592 | |
| Middle | 13.1 | 193,390 | | Drinking water treatment | | | 23.9 |
| Fourth | 11.9 | 193,370 | | No | 13.4 | 735,838 | |
| Highest | 10.5 | 193,410 | | Yes | 13.1 | 314,282 | |
| Mother's education | | | 0 | Drought indicator | | | |
| No education | 15.5 | 519,008 | | 6-month | | | 4.1 |
| Primary | 16.0 | 421,145 | | No drought | 14.1 | 888,223 | |
| Secondary | 12.0 | 366,232 | | Mild | 14.9 | 306,499 | |
| Higher | 9.2 | 73,143 | | Severe | 15.3 | 127,821 | |
| Breastfeeding | | | 29.3 | 12-month | | | 4.1 |
| No | 11.3 | 210,014 | | No drought | 14.7 | 870,147 | |
| Yes | 17.2 | 765,229 | | Mild | 14.7 | 341,003 | |
| Climate zone | | | 1.8 | Severe | 14.3 | 111,350 | |
| Tropical | 14.5 | 748,720 | | 18-month | | | 4.1 |
| Dry | 15.9 | 259,197 | | No drought | 14.8 | 916,897 | |
| Temperate | 13.0 | 336,870 | | Mild | 14.8 | 304,944 | |
| Continental | 11.0 | 2424 | | Severe | 14.6 | 100,700 | |
| Polar | 14.6 | 6921 | | 24-month | | | 4.1 |
| Source of drinking water | | | 4.6 | No drought | 14.9 | 901,347 | |
| Improved | 13.5 | 445,933 | | Mild | 14.9 | 311,287 | |
| Unimproved | 14.8 | 870,374 | | Severe | 14.6 | 109,911 | |

Incidence rate in the latest survey (%)

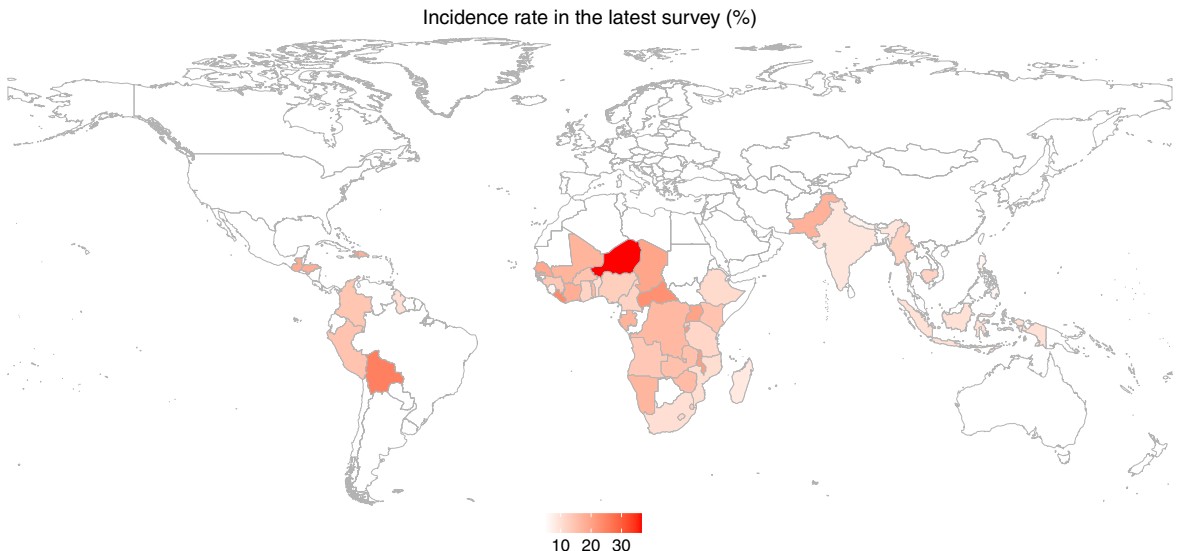

10  20  30

**Fig. 1 Incidence rate of diarrhea among children under age five in the latest survey in 51 countries during 1990–2019.**

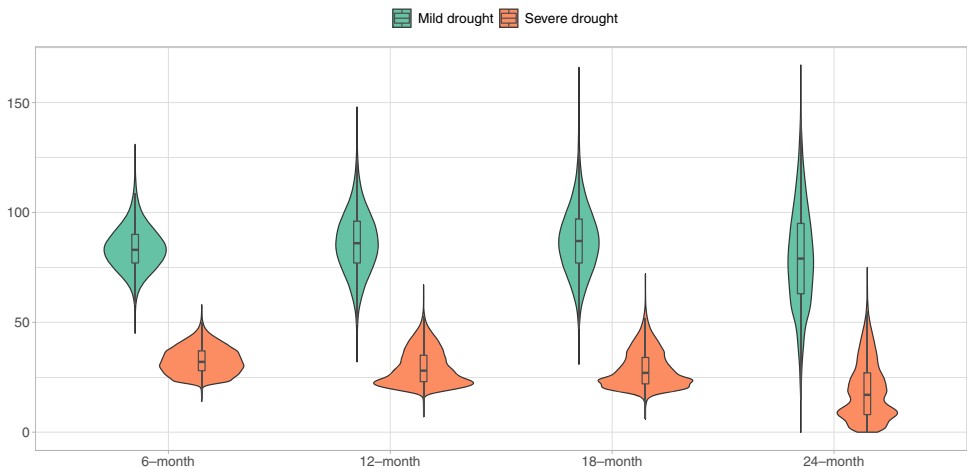

**Fig. 2 Distribution of the number of drought-months at different timescales across all 10-km-grids in 51 countries during 1990–2019.** Source data are provided as a Source Data file. The violin plots include the median values (center lines), third and first quartiles (box limits), 1.5x the interquartile range (whiskers), and the kernel probability density.

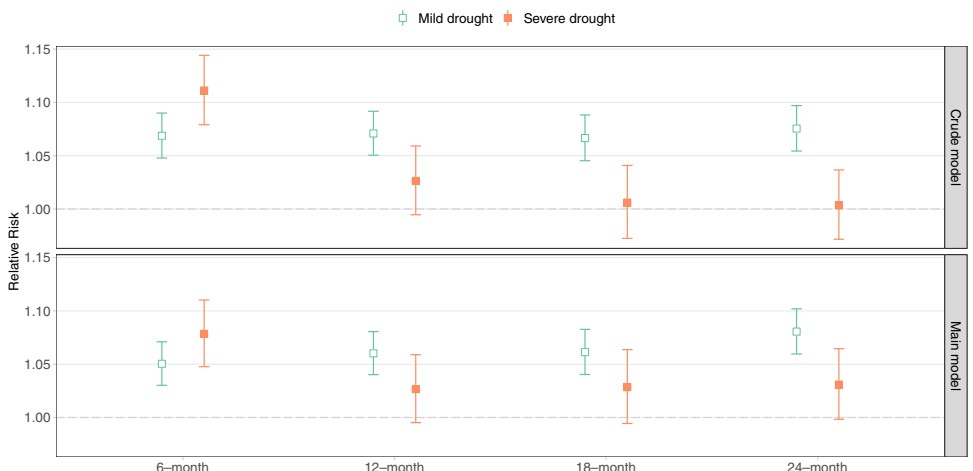

**Fig. 3 Associations between diarrhea risk among children and drought at different timescales in the crude (only adjusted for meteorological parameters, seasonality, and long-term trend) and main (further adjusted for baseline characteristics) models.** The crude and main models used the same set of observations (the set with no missing values for all variables included in the main model, $N = 713,918$). Generalized linear mixed effect models were used and no adjustments were made for multiple comparisons. Data were presented as mean values ± 1.96 × standard error.

shortest timescale of 6 months. In the crude models, the relative risks (RR) for 6-month mild and severe drought compared to no drought were 1.07 (95% confidence interval (CI): 1.05–1.09) and 1.11 (95% CI: 1.08–1.14), respectively. When assessed with the longer timescales (12, 18, or 24 months), the associations for severe drought were attenuated and not significant. In the main models, weaker associations for both mild and severe drought at 6 months were found, with RRs of 1.05 (95% CI: 1.03–1.07) and 1.08 (95% CI: 1.05–1.11), respectively. At the longer timescales, the association for severe drought was stronger than in the crude models, but at most only borderline significant (RR 1.03, 95% CI: 1.00–1.06 at the 24-month timescale).

The differences of risk estimates between the crude and main models were statistically significant for both mild and severe drought at 6 months and for severe drought at 24 months. Thus, the main model, adjusting for individual and household baseline characteristics, was used in the following analyses. We also ran another set of crude models with the full dataset (Supplementary Table 3). We observed weaker associations in the models using the larger sample, although the results showed broad consistency in terms of statistical significance.

Because we observed the strongest and only significant association for severe drought at the timescale of 6 months and observed generally consistent associations across timescales for mild drought, we conducted the mediation analysis only for the timescale of 6 months. Due to missing values for WASH variables, the sample used for this analysis was reduced to 365,975 observations (52 surveys in 35 countries) from the 713,918 observations (80 surveys in 43 countries) used for the main models. We found that 11.6% of the association between 6-month mild drought and diarrhea risk and 19.5% of the association between 6-month severe drought and diarrhea risk was mediated by WASH variables (Table 2).

We found climate zone, round-trip time to collect water, water availability at the handwashing site, and soap/detergent availability at the handwashing site to be the most robust modifiers of the association between drought and diarrhea risk (Fig. 4). We observed significant effect modification by climate zone for both mild and severe drought at every timescale (Fig. 4a). In dry climate zones, we found 24-month mild and severe drought to be strongly associated with diarrhea risk (RR: 1.22, 95% CI: 1.16–1.28 for mild drought; RR: 1.18, 95% CI: 1.09–1.27 for

**Table 2 Mediation of association between 6-month drought and diarrhea risk in children by water, sanitation, and hygiene practices.**

|  | Total effect (95% CI) | Direct effect (95% CI) | Indirect effect (Empirical 95% CI) | Percent mediated |
|---|---|---|---|---|
| Mild drought | 1.058 (1.028–1.088) | 1.051 (1.022–1.081) | 1.007 (0.996–1.015) | 11.6% |
| Severe drought | 1.135 (1.088–1.184) | 1.107 (1.062–1.155) | 1.025 (1.011–1.033) | 19.5% |

Empirical 95% CI: 95% confidence interval for indirect effect generated from 1000 bootstrap samples; percent mediated: proportion of association between drought and diarrhea mediated by water, sanitation, and hygiene variables, calculated as the difference between the drought exposure coefficient from the main model (total effect; adjusted for meteorological parameters, seasonality, long-term trend, and baseline characteristics) and the drought exposure coefficient from the WASH-adjusted model (direct effect; further adjusted for WASH variables) divided by the drought exposure coefficient from the main model. The main and adjusted models used the same set of observations, with no missing values for all variables included in the adjusted model (N = 365,975).

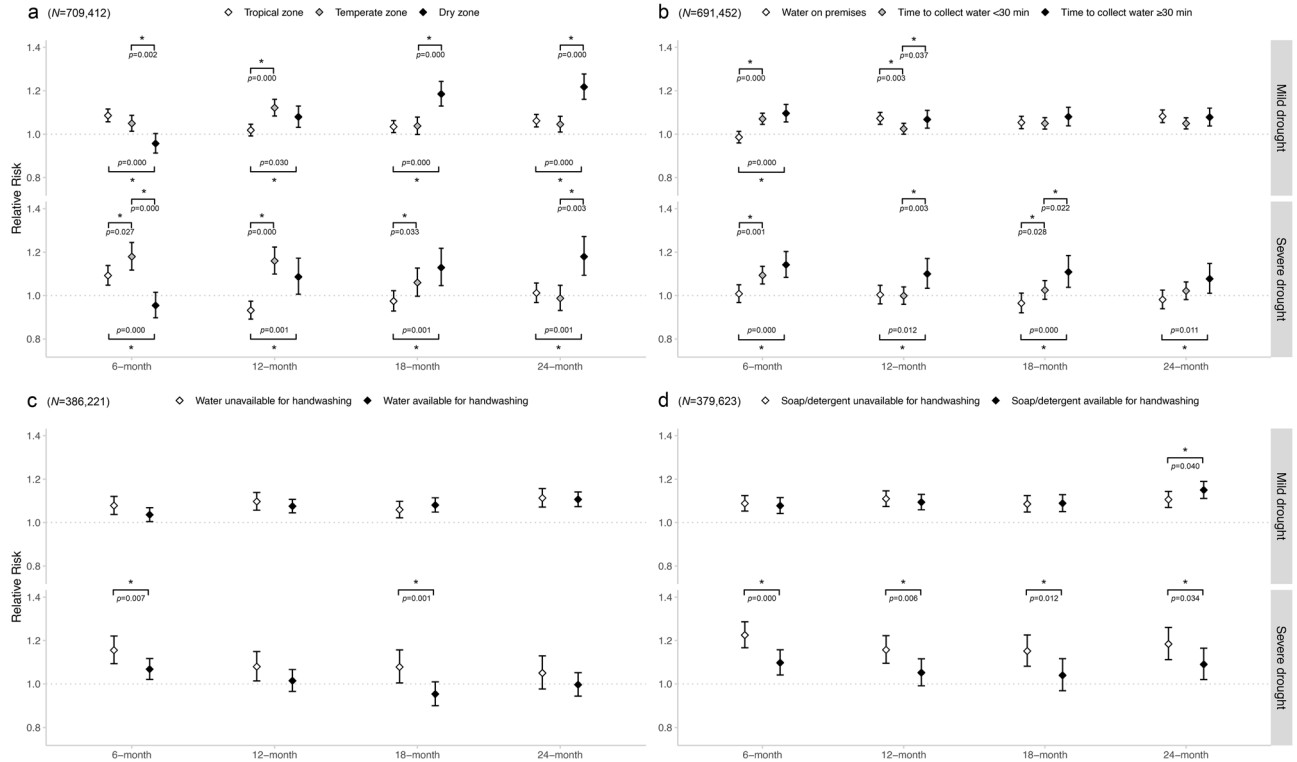

**Fig. 4 Associations between diarrhea among children and drought stratified by climate zone (a), round-trip time to collect water (b), water availability at handwashing site (c), and soap/detergent availability at handwashing site (d).** The sample size was 709,412, 691,452, 386,221, and 379,623 children for stratified analysis for climate zone (**a**), round-trip time to collect water (**b**), water availability at handwashing site (**c**), and soap/detergent availability at handwashing site (**d**), respectively. Statistically significant pairwise differences are marked with an asterisk. Generalized linear mixed effect models were used and no adjustments were made for multiple comparisons. Data were presented as mean values ± 1.96 × standard error. The p values are two-sided.

severe drought), whereas in tropical or temperate zones these associations were weak (mild drought) or null (severe drought). This pattern was reversed for a 6-month mild and severe drought, with elevated RRs in tropical and temperate zones but not in dry zones. Furthermore, for both mild and severe drought, RRs in dry zones increased with increasing timescale length in a graded fashion. However, RRs in temperate zones decreased with increasing timescale length in a graded fashion for severe drought, with no apparent trend for tropical zones across timescales.

Round-trip time to collect water, water availability at the handwashing site, and soap/detergent availability at the handwashing site each modified the association between severe drought and diarrhea risk in a consistent pattern across timescales. Specifically, for each timescale, the RR was elevated among children living in households that needed ≥30 min to collect water but was null among children living in households with water on premises (Fig. 4b). Among children living in

households that needed ≤30 min to collect water, the association was null for all timescales except 6 months. The association also was stronger among children with no water or soap/detergent available at their handwashing site than among children with water or soap/detergent available, respectively (Fig. 4c, d). Results for effect modification by other baseline characteristics and WASH variables are shown in Supplementary Figs. 5–14.

**Sensitivity analysis**. The sensitivity analysis showed our results to be robust, with associations generally in agreement with our findings in the main analysis (Supplementary Figs. 15–19). The analyses using alternative sources of SPEI with coarser spatial resolution found significantly elevated diarrhea risk for both mild and severe drought at the 6-month timescale, with a higher RR for severe drought, consistent with the main analysis (Supplementary Figs. 15, 16 and Fig. 3). However, using these alternative SPEI sources, the association between severe drought and diarrhea risk was significant at all timescales, whereas in the main

analysis this association was significant only at the 6-month timescale. Results using multiple imputations were highly consistent with the main analysis (Supplementary Fig. 17). After disaggregating the drought variable, the strongest association between drought and diarrhea risk was observed for extreme or exceptional drought (SPEI $\leq -1.6$) at the timescale of 6 months (RR: 1.14, 95% CI: 1.09–1.18); however, this association was null at the other timescales. Significant associations between diarrhea risk and the less extreme drought categories were observed at all timescales, with the strongest association observed for 18- and 24-month moderate drought (Supplementary Fig. 18). After the inclusion of nutrition indicators (as individual covariates in separate models), the estimates for both mild and severe drought at all timescales did not appreciably change (Supplementary Fig. 19). Finally, similar to our main mediation analysis, our alternative mediation analysis (with a different set of model specifications) indicated that a low to moderate proportion of the overall effect of drought (0.7–27.0% for mild drought and 1.0–30.6% for severe drought) was mediated by WASH practices. However, these proportions represent the range for individual WASH practices (Supplementary Table 4), whereas the main mediation analysis assessed the proportion mediated by all WASH practices combined.

## Discussion

Using an advanced measure of drought, we observed associations between exposure to long-term drought and elevated diarrhea risk among children under age five in LMICs. We observed significantly positive associations for mild drought across timescales (6 to 24 months) and for severe drought at the 6-month timescale, with borderline significance associations for severe drought at the other timescales. At the 6-month timescale, WASH practices mediated a low proportion of the association with mild drought and a moderate proportion of the association with severe drought.

The literature on the relationship between drought and diarrhea is very limited. Some previous studies found a positive relationship between the risk of diarrhea and low rainfall[11–13,17,18], which was employed by a few of these studies as the drought indicator[17,18]. Some other studies have used indirect surrogates to represent drought exposure. For example, an early study reported a dose-response relationship between drought-induced water supply deficiencies and the incidence of diarrhea or vomiting in children[27]. However, our study used a more advanced measure of drought that took into account both water demand and water supply. As mentioned, we are aware of only one other study that used a complex measure of drought; this study found an association between aridity and elevated diarrhea incidence in Afghanistan[22].

Decreases in quality or quantity of surface water or groundwater may increase the risk of diarrheal disease. Drought, through reduced precipitation and increased evapotranspiration, causes declines in surface water and groundwater levels, raising the concentration of diarrheagenic agents[28]. The limited water supply could provoke intensified groundwater pumping, particularly for irrigation in agricultural areas[29], which could further contribute to groundwater depletion. Nutrient-load-associated eutrophication of water bodies, induced by water deficits, could stimulate the increased growth of enteric pathogens[30]. Drought, accompanied by increased pumping of groundwater, could result in saltwater intrusion into surface water or groundwater, making these water sources unusable due to high salinity[31]. Moreover, during drought, there also is the possibility of piped/tap water contamination due to increased turbidity from the loss of pressure at water treatment plants[30], and epidemiological evidence has demonstrated the association between gastrointestinal illness and water distribution system deficiencies during periods of interrupted supply[32].

In these ways, drought and its various adverse effects on water quantity and quality could cause an elevated risk of diarrheal disease among children regardless of water source, including domestic wells, water tanks, and piped/tap water. In addition, treated municipal sewage may be used for irrigation of crops during periods of water shortage[30], leading to increased diarrhea risk from food contamination. Finally, chronically impaired local physical infrastructure for water storage and treatment available to economically disadvantaged households in LMICs, as well as insufficient financial capacity for adaptation during droughts, pose public health challenges for infectious diarrheal diseases in LMICs.

We assessed the plausibility of the mediating pathway through WASH and found that a low to moderate portion of the association between drought and diarrhea risk was mediated by WASH practices. This showed that WASH practices can only partially avert the impacts of climate change-induced drought. However, the traditional method we used for the mediation analysis is prone to bias with the presence of interaction between the exposure (i.e., drought) and mediator (i.e., WASH practices)[33]. Recently developed methods employing a counterfactual framework either do not accommodate multiple mediators[34] or are incapable of taking high-dimensional interactions (seven-way interaction in this study) into account[35]. Considering the study objective to explore the mediating nature of WASH, we ultimately adopted the traditional method, acknowledging the fundamental limitation in interpreting our mediation analysis results.

For severe drought across all timescales, our effect modification analysis revealed consistently stronger associations among children in households with a longer time to collect water or without water or soap/detergent for handwashing. Under drought conditions, households with a longer time to collect water may be unable to collect adequate amounts of water to meet their children's routine hygienic needs. Similarly, under drought conditions, handwashing among children without available water or soap/detergent may be inadequate to prevent diarrhea, given enhanced enteric pathogen exposure. Thus, children in households with greater WASH capacities appeared to be less vulnerable to increased diarrhea risk during drought.

In dry climate zones, we found both mild and severe drought to be strongly associated with diarrhea risk at the 24-month timescale, with more modest associations at the 12- and 18-month timescales and null associations at the 6-month timescale. An explanation for this pattern could be that dry regions, being already dry, have learned to adapt to drought in the shorter term (i.e., 6 months), but do not have the capacity to adapt to more persistent drought conditions. In contrast, in tropical and temperate climate zones, we found both mild and severe drought to be associated with diarrhea risk at the 6-month timescale but found null associations for severe drought at the 24-month timescale. Thus, tropical and temperate zones may have less capacity to adapt to drought than already dry zones in the shorter term, with greater resources for adaptation in the longer term.

The strong association between prolonged drought and diarrhea risk in dry climates raises particular concerns in settings that already suffer from water scarcity. Climate change-induced shifts in precipitation patterns, in which wet regions generally are expected to become wetter and dry regions generally are expected to become drier[36], are likely to lead to longer drought durations in dry climate zones, with resultant increased risk of diarrhea in children under age five. Furthermore, climate change may lead to increased frequency, duration, and intensity of 6-month drought

in tropical and temperate regions[37], potentially increasing the risk of diarrhea in children under age five living in these climate zones as well.

We observed effect modification by some baseline characteristics and WASH variables to be inconsistent across timescales or between mild versus severe drought. It is possible that the modifying effects of these factors differ across local policy contexts and socioeconomic development. It is also possible that some of these inconsistencies were due to random variability. The reproducibility of these inconsistent findings needs further investigation.

Other anthropogenic factors that this study did not assess but could potentially modify the association investigated include food safety and population migration. As discussed above, drought can lead to the use of contaminated water for irrigation due to clean water deficiency, with resultant food contamination. In addition, drought favors the growth of mycotoxin-producing fungi on cereal, nuts, and other crops[38–40]. Ingestion of trichothecenes, in particular, which are produced by *Fusarium* fungi, may lead to diarrheal illness[41]. Moreover, post-drought heavy rainfall causes severe runoff that may increase the risk of pathogen diffusion, further worsening soil and water contamination[39].

The linkage between drought and migration and food insecurity has also been investigated in previous studies[42,43]. Under climate change, a more frequent drought would exacerbate land degradation[42], reduce agricultural production, and increase rural unemployment, thus aggravating household food insecurity[43] and potentially increasing the risk of malnutrition and diarrhea. Migration in response to deteriorating rural livelihood may serve as an adaptation strategy that has been found to improve food security[43], although the choice of this drought-related migration may be conditioned on local economic, political, and social context[42]. However, forced migration poses a threat of food and water insecurity[39] that could increase diarrhea risk. Future studies should take the complex interrelationship between climate change, food safety and security, and migration into account when investigating the association between drought and health outcomes.

This is so far the largest study investigating the impact of long-term drought on the risk of diarrhea among children in LMICs, employing a nationally representative multi-country dataset for ascertainment of diarrhea cases and an advanced measure of drought (i.e, SPEI), at a high spatial resolution to minimize possible exposure misclassification. Nonetheless, several limitations need to be acknowledged. First, we assigned 10-km gridded exposure to drought to each surveyed child in a specific cluster. However, the geocoded sites were randomly displaced for 2–10 km based on the area of residence to protect respondent anonymity[44]. Although our spatial resolution was able to address most of the displacement, there was still the chance of mismatching the exposure with a survey cluster, especially for the clusters close to a boarder of a 10-km grid. Second, currently, no universal definition for drought is available, and there are 50 indicators and indices listed by the Integrated Drought Management Program from the World Meteorological Organization[45]. Moreover, the representation of drought severity by SPEI or SPI is based on relatively arbitrary classifications that vary across authorities and locations[46–48]. Therefore, our findings may not be generalizable, although we included data from sites in 51 countries across a broad geographical range.

Third, due to missing values for baseline characteristics (primarily age and wealth index), the sample used for our main model included only 713,918 children, reduced from 1,379,566 children in the original dataset. Furthermore, due to missing values for WASH variables, the sample used for our mediation analysis included only 365,975 children. The use of these reduced sample sizes introduced the possibility of selection bias, and discrepancies in RRs between models using different sample sizes were indeed found. Nevertheless, the direction and significance of associations across models remained largely consistent (i.e., Supplementary Table 3). Fourth, the health outcome, whether each child under five had diarrhea in the previous two weeks before the interview, was reported by the mother and is susceptible to inaccurate recall leading to misclassification of diarrhea status. Fifth, we tested possible effect modification by a set of 14 variables for two types of drought (mild and severe) at four timescales, introducing the possibility of chance associations due to multiple comparisons. However, in our interpretation of effect modification results, we emphasized consistency across timescale within each drought type (mild or severe). In addition, we consider the effect modification analysis to be an exploratory analysis that requires replication.

Sixth, diarrhea cases were not diagnosed using laboratory tests, meaning that no information on the causative agent was available. We, therefore, were unable to study the association between drought and the risk of diarrhea caused by different pathogens (viruses, bacteria, or parasites), which have distinct biological characteristics and etiological mechanisms for causing diarrhea[49]. Lastly, in addition to food safety, food insecurity, and migration discussed above, there are other factors by which diarrhea burdens could vary considerably, especially water supply interruption patterns in slums. Our analysis could not disentangle the interrelationship between drought and all these factors due to data unavailability.

In conclusion, we found that exposure to long-term drought was associated with higher diarrhea incidence among children under the age of five years in LMICs and that this association was modified by climate zone, round-trip time to collect water, and water or soap/detergent availability for handwashing. We also observed that WASH practices mediated a low to moderate proportion of the association between long-term drought and the risk of diarrhea. Given the inadequate capacity to cope with and adapt to extreme weather events such as drought in LMICs[50], and with drought expected to intensify in already vulnerable regions due to climate change[2], intergovernmental collaborative actions are urgently needed to alleviate the public health burden from droughts, particularly by improving WASH infrastructure among socioeconomically disadvantaged populations[51].

## Methods

**Diarrhea data.** We obtained data on diarrhea incidence among children under five and on other individuals/household variables from the Demographic and Health Surveys (DHS)[52]. Since 1984, the DHS program has conducted nationally representative household surveys in LMICs, covering a wide range of topics such as family planning, maternal and child health, malaria, nutrition, and environmental health. In this study, we included the 141 surveys, conducted in 51 LMICs, that had a corresponding global positioning system dataset available at the time we received the data in August 2021. In the DHS sample design, clusters were randomly selected from a list of census enumeration areas stratified by geographic region and by urban/rural area within each region. Households were then randomly selected from each cluster. The geographic center of each selected cluster was calculated and used as the coordinate location for each selected household located within the cluster[53].

Our measure of diarrhea incidence, whether each child under age five living in the household had had diarrhea in the two weeks before the interview, was obtained from a woman of reproductive age (15–49 years) living in the household. Sociodemographic and behavioral variables, including child's age and sex, mother's education, rural/urban area of residence, and history of breastfeeding were also reported by the mother, while WASH variables, including a source of drinking water, round-trip time to collect water, drinking water treatment, a place to wash hands, availability of water, or soap/detergent, at handwashing site, type of toilet facility, and variables used to calculate household wealth index were reported by any knowledgeable person aged 15–59 years or older living in the household.

DHS surveys query about common household assets and services and use this information to calculate a household wealth index within each survey, standardized to the survey-specific asset distribution. However, this wealth index does not allow

comparisons across surveys, given differences in absolute wealth levels across countries and calendar time. To address this problem, we used a method described by Bendavid[54], in which we pooled all households with information on the following assets and services: source of drinking water, toilet facilities, electricity, type of flooring, number of rooms per person living in the household, and possession of radio, television, landline, cellphone, refrigerator, motorcycle, and car. We then generated a wealth index using principal component analysis and used the quintiles of this index in our analyses. The index was weighted by the ratio of the country population to the number of surveyed children in a specific survey year to avoid undue influence from uniquely large surveys[54].

**Drought exposure**. We measured drought using the SPEI[21,26], a recently developed index with advantages over two widely applied drought indicators, the Palmer drought severity index and standardized precipitation index (SPI). SPEI combines the sensitivity of the former to changes in evaporation demand caused by temperature with the multi-scale nature of the latter[21]. It calculates the climatic water balance by comparing the available water content of soil and vegetation with the atmospheric evaporative demand and thus provides a more reliable measure of drought severity in a warming world than only considering precipitation[21].

We first downloaded and rescaled monthly averaged meteorological records at a resolution of 0.1° (~10 × 10 km) during 1988–2019 from the fifth-generation European Centre for Medium-Range Weather Forecasts atmospheric reanalysis of the global climate (ERA5-Land)[55], including mean temperature, dewpoint temperature, total precipitation, wind speed, atmospheric pressure, and incoming solar radiation. Monthly mean daily maximum and minimum temperatures of the same period were calculated using hourly air temperature records downloaded from ERA5-Land. Global terrain elevation data at a resolution of 0.01° were obtained from the Global Multi-resolution Terrain Elevation Data 2010[56].

Next, we calculated the monthly potential evapotranspiration (PET) using the FAO-56 Penman-Monteith equation[57], which incorporates maximum and minimum temperatures, dewpoint temperature, wind speed, solar radiation, air pressure, latitude, and elevation. It is a more accurate, robust, and recommended method over the Hargreaves and Thornthwaite equation[58,59], given the data required are available. Using PET and monthly total precipitation, we then calculated the $10 \times 10$ km SPEI at different timescales (i.e., 6/12/18/24 months) for each month during 1990–2019. These timescales denote arbitrary but typical timescales of drought[60], accounting for the cumulative effects of climate conditions during the corresponding antecedent months as a whole[21]. For example, to calculate the 6-month SPEI of a specific month, a time-series is constructed by the sum of water balance (precipitation minus PET) from five months before to the current month; then a normalization process is performed to this series to remove the effect of both season and climate regime of each site[59]. In addition, the SPEIs of different timescales are independent of each other due to the cumulative nature of the calculation[59]. Similar to SPI, droughts at shorter timescales (<6 months) are mainly related to soil moisture deficits, whereas droughts at longer timescales (≥6 months) are related to variations in the reservoir and groundwater storages[47,61]. It has also been found that 1–24 months is the best practical range application for SPI, and timescale intervals longer than 24 months may be unreliable[47,62]. Therefore, only the SPEI at scales of 6–24 months were considered in our analysis.

We then linked the gridded monthly SPEI and 2-month moving averages (lag 0–1 month) of mean temperature and total precipitation with each child under age five in all surveys based on the geographic location of each survey site and the month and year when the mother was interviewed. Finally, we classified drought events based on their severity according to the definition from the Federal Office of Meteorology and Climatology MeteoSwiss[46]. We combined mild and moderate drought into a category we termed mild drought ($-1.3 <$ SPEI $\leq -0.5$) and combined severe, extreme, and exceptional drought into a category we termed severe drought (SPEI $\leq -1.3$) to ensure an adequate sample size in each category. Thus, we constructed two drought categories corresponding to each of the four timescales, and each month/year of each survey site was classified as no drought, mild drought, or severe drought for each of the timescales of 6 months, 12 months, 18 months, or 24 months, respectively.

**Analytical approach**. All data analyses were completed using R statistical software 4.0.2, with the *SPEI* package for drought exposure assessment, and the *MASS* package for regression analysis.

**Main models**. Drought-timescale-specific log-binomial generalized linear mixed models were used to regress the risk of having diarrhea in the previous two weeks on drought occurrence:

$$\mathrm{Log}(P_i) = \beta_0 + \beta_1 X_j + \beta_2 \mathrm{COV}_i + \varphi_j + \lambda + \omega_j$$

where $i$ is the index for each individual child and $j$ is the DHS survey site in a specific country at a specific time, $P_i$ is the probability of having diarrhea. $X_j$ is the drought indicator (no drought, mild, or severe drought) at a specific timescale for a specific site and survey month. $\mathrm{COV}_i$ is a matrix of individual and household baseline characteristics, including age, sex, mother's education, area of residence, and wealth index. Previously found to be associated with diarrhea in children[63],

2-month averaged mean temperature and total precipitation ($\varphi_j$) were included in the models as a matrix of natural cubic splines with three degrees of freedom. $\lambda$ is a matrix of natural cubic splines of survey month and year, both with three degrees of freedom, to adjust for seasonality and long-term trends, respectively. $\omega_j$ denotes the random effect for survey cluster, accounting for cross-cluster differences. To examine the need to adjust for individual and household baseline characteristics, we compared two separate models using the same set of observations – the crude model (without baseline characteristics) and the main model (with baseline characteristics). All results are reported as the RR associated with exposure to drought compared to non-exposure to drought, along with 95% CIs. The statistical significance of the difference between estimates of two models was tested by calculating the 95% CI as $(\hat{Q}_1 - \hat{Q}_2) \pm 1.96\sqrt{\left(\widehat{SE}_1\right)^2 + \left(\widehat{SE}_2\right)^2}$, where $\hat{Q}_1$ and $\hat{Q}_2$ are the estimates for different models, and $\widehat{SE}_1$ and $\widehat{SE}_2$ are their respective standard errors[64].

**Mediation analysis framework**. We used mediation analysis to examine the degree to which WASH variables mediated the association between drought and diarrhea risk. Specifically, to estimate the indirect effect of drought exposure, we separately fit two models—the main model (adjusted for baseline characteristics) and the WASH-adjusted model (adjusted for both baseline characteristics and WASH variables)—for the same complete dataset (without missing values for any of the baseline characteristics or WASH variables). We calculated the association between drought and diarrhea mediated by WASH variables (the indirect effect of drought) as the drought exposure coefficient in the main model (the total effect) minus the drought exposure coefficient in the WASH-adjusted model (the direct effect)[65]. We then obtained the proportion of the effect mediated by WASH by dividing the indirect effect by the total effect from the main model. We calculated empirical 95% CIs for the indirect effect by generating 1000 bootstrap samples[66]. Logistic models were undesirable in our study because the outcome was not rare (Table 1), and thus the odds ratios would have overestimated the risk ratios. Moreover, a common outcome would invalidate traditional approaches for mediation analysis; one workaround for this issue is to estimate the direct and indirect effects on a risk ratio scale using a log-binomial model[65].

**Effect modification analysis**. We examined possible effect modification by adding to the main model (adjusted for baseline characteristics) an interaction term (or the main effect and an interaction term for variables not in the main model) between drought and each of the following factors: age (<24 months and 24–59 months), sex, mother's education (primary or lower and secondary or higher), area of residence (urban and rural), wealth index (1st–3rd quintile and 4th–5th quintile), breastfeeding history (no and yes), Köppen climate classification (tropical, dry, temperate, continental, and polar), and the following WASH variables: source of drinking water (improved and unimproved), round-trip time to collect water (water on premises, <30 min, and ≥30 min), water treatment for drinking (no and yes), a place to wash hands (fixed and mobile places), availability of water, or soap/detergent, at handwashing site (no and yes), and type of toilet facility (improved and unimproved).

**Sensitivity analysis**. We tested the robustness of our results in several ways, using the main model (adjusting for baseline characteristics). First, we used two additional data sources for the representation of drought events. Specifically, we downloaded the SPEI at coarser spatial resolutions provided by the Global SPEI database[61] and by the Earth Observation for Sustainable Development from the European Space Agency[67] and matched them with each surveyed child. The former source calculated SPEI using the Penman-Monteith equation with a 0.5° (~50 × 50 km) spatial resolution and the latter used the Thornthwaite equation with a 0.25° (~25 × 25 km) resolution. For these data sources, drought was defined in the same way as in the main analysis.

Second, we conducted multiple imputations ($n = 5$) to fill in each missing value, then re-performed the regression analysis for each imputed dataset separately, and finally pooled the results together. This procedure was implemented by multivariate imputation by chained equations (MICE)[68]. Although MICE can handle both data missing at random and not missing at random, multiple imputations under the latter scenario requires additional modeling assumptions that could influence the generated imputations[68]. WASH variables were likely to cause a systematic missing issue (missing simultaneously) in a survey, and thus we only implemented the imputation for the variables included in the main models.

Third, we disaggregated the drought variable into four categories instead of two (i.e., $-0.8 <$ SPEI $\leq -0.5$ as mild drought, $-1.3 <$ SPEI $\leq -0.8$ as moderate drought, $-1.6 <$ SPEI $\leq -1.3$ as severe drought, and SPEI $\leq -1.6$ as extreme/exceptional drought). We continued to combine extreme and exceptional drought into a single category to avoid excessively unbalanced sample sizes across categories.

Fourth, we included the nutrition status of the child as a covariate to test its influence on our estimates, including height-for-age z-score (a measure for stunting), weight-for-age z-score (a measure for wasting), and weight-for-height z-score (a measure for underweight). Specifically, linear terms of these three indicators were incorporated into separate models.

Fifth, we used an alternative method to examine the mediating effect of WASH variables, allowing for interaction between drought and these variables, in which

the overall effect of drought can be decomposed into the proportion of the effect due to (1) the direct effect of drought; (2) the interaction between drought and WASH alone; (3) both mediation by WASH and the aforementioned interaction; (4) mediation by WASH alone[34]. However, this method does not allow non-linear covariates, random effects, or multiple mediators, so we included mean temperature, rainfall, and survey year as linear terms, removed the random effect variable of survey sites, and assessed the mediating effect of each WASH variable in a separate model. Because WASH variables potentially affect one another, their individual mediation proportions cannot be summed up for an overall mediated effect for all WASH variables.

**Reporting summary**. Further information on research design is available in the Nature Research Reporting Summary linked to this article.

## Data availability

Survey data including diarrhea and socioeconomic data in this study are publicly available upon request from the Demographic and Health Surveys Program (https://dhsprogram.com/). Publicly available meteorological records can be downloaded from the fifth-generation ECMWF atmospheric reanalysis of the global climate (ERA5-Land) at https://cds.climate.copernicus.eu/. The SPEI at a resolution of 0.5° (~50 × 50 km) calculated by the Global SPEI database can be downloaded at https://spei.csic.es/spei_database. The SPEI at a resolution of 0.25° (~25 × 25 km) calculated by the European Space Agency can be downloaded at https://explorer-eo4sdcr.adamplatform.eu/. Source data on the calculated SPEI for the regression models and data on drought-month by drought severity and timescale are provided with this paper at https://github.com/CHENlab-Yale/drought-diarrhea or https://doi.org/10.5281/zenodo.6527455[69].

## Code availability

The programming code for the main models and mediation analysis is available at https://github.com/CHENlab-Yale/drought-diarrhea or https://doi.org/10.5281/zenodo.6527455[69].

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

## Acknowledgements

The authors would like to thank the DHS Program and ICF International for providing the data used in the analysis. This work was funded by the Reckitt Global Hygiene Institute (RGHI RIN: 2021-001). The views expressed are those of the authors and not necessarily those of RGHI. The funders had no role in considering the study design or in the collection, analysis, interpretation of data, writing of the report, or decision to submit the article for publication. V.E.P. was supported by grant R01AI112970 from the U.S. National Institutes of Health/National Institute of Allergy and Infectious Diseases. R.D. received support from the High Tide Foundation.

## Author contributions

K.C., R.D., and P.W. conceived and designed the study and developed the statistical analysis strategy. P.W. and K.C. have directly accessed and verified the underlying data. P.W. prepared and cleaned the data, wrote the scripts for the statistical analyses, performed the exposure assessment and statistical analysis, and wrote the original draft of the manuscript. K.C., E.A., R.D., and V.E.P. contributed to the interpretation of the results, reviewed the manuscript, and edited the original version. K.C. and R.D. contributed equally as joint last authors. All authors have approved the final draft of the manuscript.

## Ethics approval

This study was determined by the Yale Institutional Review Board as not-human-subject research (IRB ID: 2000030174) and thus ethics approval was not required for the current study.

## Competing interests

V.E.P. has received reimbursement from Merck and Pfizer for travel to Scientific Input Engagements unrelated to the topic of this study and is a member of the WHO Immunization and Vaccine-related Implementation Research Advisory Committee (IVIR-AC). The remaining authors declare no competing interests.
