## [Peer Review File · Nature Communications]

Reviewers' Comments:

Reviewer #1:

Remarks to the Author:

This is an important and robust analysis linking available information related to drought and a standardized precipitation evapotranspiration index (SPEI) and finds robust associations between prolonged and severe drought conditions and childhood diarrhea in 51 LMIC countries. The authors have conducted fairly robust analyses and describe it clearly.

However, several aspects of the project could do with greater clarity. one of those is the choice of countries and surveys. why were countries with available DHS surveys excluded?

An important interface of climate change, draught and population migration isn't captured. How did these surveys capture elements related to that connection which could be a driver of disease and ill health as well. It wasn't clear if nutritional status of index child was taken into account in determining risks of diarrhea.

The authors did take breastfeeding into account in adjusting risks but the other pathway to diarrhea, beyond safe water, is safe foods. Dietary intake data may have been sparse but the authors should comment on how the food environment and hence food safety was taken into account among children older than 6 months of age?

I know that the authors took WASH into account and looked at risks by various patterns. Did they adjust for type of water supply and patterns by seasonality. Diarrhea and enteric infection burdens vary considerably by season, water supply interruption patterns in slums, periodic flooding etc. Were these investigated?

Reviewer #2:

Remarks to the Author:

In this study, Wang et al investigate the relationship between drought and self-report diarrhea in 51 countries using DHS data. The authors generally found that drought (measured using the SPEI) was associated with increased diarrhea, but that the time scale at which this effect operated depended on the climate zone and was also modified by local WASH infrastructure, with areas that had poor access to WASH services being more vulnerable to the effects of drought. Overall, this is an important paper that fills a key gap in the field regarding the relationship between drought and diarrhea and I read the work with great interest. The inclusion of multiple countries in this analysis greatly strengthens the conclusions that can be drawn from the paper. While there are issues inherent with using DHS data, the authors have pointed out the most critical issues in the discussion. While multiple comparisons were used, the overall high level of consistency across contexts and with the different sensitivity analyses strengthens my confidence int the authors' main findings. However, I am not certain that the approach used to address mediation is valid, particularly in the context of effect modification.

My main concern with the analysis is that the authors used the method of adjustment to look at mediation, which has many known weaknesses (see Richiardi et al, 2013, IJE for a summary), particularly because these variables might also be effect modifiers. The authors might want to consider an alternative method that considers interaction and mediation jointly in a more formal way. For example, the approach described by Vanderweele in doi: 10.1097/EDE.000000000000121 could be used here. If the authors feel that the method of adjustment is justified in this case, please provide further justification in the main text. Relatedly, I recommend that the authors move some of the discussion text about mediation vs. effect modification to the introduction section to introduce more fully the reason why these WASH variables might function in both ways, which I agree is possible. Adding some text to the introduction will help the reader to more easily follow the paper narrative.

Additionally, as a more minor comment, I wonder if the authors are adequately powered to look at the impact of 24-month droughts in tropical climates specifically. Could the authors provide some

data (in the SI would be fine) about the number of drought months with data by climate zone? Figures S2-S4 are helpful, but because climate zone is not shown on the map I cannot be sure if the authors are adequately powered to test this interaction.

Response to the editor and reviewers

We sincerely thank all reviewers for the comments and suggestions that significantly improve this manuscript. We made major revisions to the previous draft, particularly for the Results and Discussion, to address the comments from the reviewers (blue colored) and marked down the corresponding location in the text for each response below.

Reviewer 1

Q1. This is an important and robust analysis linking available information related to drought and a standardized precipitation evapotranspiration index (SPEI) and finds robust associations between prolonged and severe drought conditions and childhood diarrhea in 51 LMIC countries. The authors have conducted fairly robust analyses and describe it clearly. However, several aspects of the project could do with greater clarity. one of those is the choice of countries and surveys. why were countries with available DHS surveys excluded?

A1. Thank you for your comment. We only included DHS surveys with both the survey dataset and GPS dataset available, with the latter used to match the estimated drought condition to each survey cluster. Up to the time we collected the data (August 2021), there were in total 141 surveys conducted in 51 low- and middle-income countries (LMICs) that were finally included in our study. We have now made this choice clearer by revising the method sentence as follows: “In this study we included the 141 surveys, conducted in 51 LMICs, that had a corresponding global positioning system dataset available at the time we received the data in August 2021.” (Lines 343-345).

Q2. An important interface of climate change, draught and population migration isn't captured. How did these surveys capture elements related to that connection which could be a driver of disease and ill health as well. It wasn't clear if nutritional status of index child was taken into account in determining risks of diarrhea.

A2. Thank you for this question/comment. Up to now there has been no information on population migration in DHS surveys. But we agree that the interaction between climate change

and migration plays an important role in the health impact of long-term drought. Regarding this, we now added a separate paragraph in the Discussion:

“Other anthropogenic factors that this study did not assess but could potentially modify the association investigated include food safety and population migration..... The linkage between drought and migration and food insecurity has also been investigated in previous studies^{1, 2}. Under climate change, more frequent drought would exacerbate land degradation¹, reduce agricultural production, and increase rural unemployment, thus aggravating household food insecurity² and potentially increasing the risk of malnutrition and diarrhea. Migration in response to deteriorating rural livelihood may serve as an adaptation strategy that has been found to improve food security², although the choice of this drought-related migration may be conditioned on local economic, political, and social context¹. However, forced migration poses a threat of food and water insecurity³ that could increase diarrhea risk. Future studies should take the complex interrelationship among climate change, food safety and security, and migration into account when investigating the association between drought and health outcomes.” (Lines 270-287).

We have also added this to the limitation section:

“Lastly, in addition to food safety, food insecurity, and migration discussed above, there are other factors by which diarrhea burdens could vary considerably, especially water supply interruption patterns in slums. Our analysis could not disentangle the interrelationship between drought and all these factors due to data unavailability.” (Lines 321-325)

For nutrition status, in the new sensitivity analyses we included three DHS nutrition indicators (height-for-age z score, weight-for-age z score, and weight-for-height z score) as covariates, each in a separate model, to test how they would influence our estimates. Please see Figure R1 below.

Figure R1. Associations between diarrhea in children and drought at different timescales after including nutrition indicators as covariates ($N=560,768$).

The estimates from the main model are slightly different from those in Figure 3 because a dataset with a different number of observations was used (713,918 for the main model in Figure 3 and 560,768 for the main model here due to additional missing values for nutrition indicators).

From this figure, after including any nutrition indicator, the changes in estimates are negligible, except that the risk associated with 12-, 18-, and 24-month severe drought slightly increased in the weight-for-age and weight-for-height models. In addition, in a paper by Cooper et al., the

authors found a significant positive impact of diarrhea on child stunting (height-for-age z score < -2.0)⁴. This suggests that a previous episode of diarrhea is also a possible cause of malnutrition, making it inappropriate to adjust for nutritional status in our study since it could introduce confounding of the association between drought and diarrhea due to other common causes of diarrhea at different time points. Therefore, considering the complex causality between diarrhea and nutrition status and the overall minor influence of these indicators on our estimates, we decided not to include nutritional indicators as covariates in our main models. However, we now incorporate this as a sensitivity analysis. We now have made revisions in the Methods and Results:

“Fourth, we included nutrition status of the child as a covariate to test its influence on our estimates, including height-for-age z score (a measure for stunting), weight-for-age z score (a measure for wasting), and weight-for-height z score (a measure for underweight). Specifically, linear terms of these three indicators were incorporated into separate models.” (Lines 488-491)

“After inclusion of nutrition indicators (as individual covariates in separate models), the estimates for both mild and severe drought at all timescales did not appreciably change (Supplementary Figure 19).” (Lines 184-186)

Q3. The authors did take breastfeeding into account in adjusting risks but the other pathway to diarrhea, beyond safe water, is safe foods. Dietary intake data may have been sparse but the authors should comment on how the food environment and hence food safety was taken into account among children older than 6 months of age?

A3. Thank you for your comment. Together with population migration, we have already added a section discussing the potential influence of food safety on the association between drought and diarrhea (lines 270-277).

“Other anthropogenic factors that this study did not assess but could potentially modify the association investigated include food safety and population migration. As discussed above, drought can lead to use of contaminated water for irrigation due to clean water deficiency, with resultant food contamination. In addition, drought favors growth of mycotoxin-producing fungi on cereal, nuts and other crops^{3,5,6}. Ingestion of trichothecenes, in particular, which are produced

by *Fusarium* fungi, may lead to diarrheal illness⁷. Moreover, post-drought heavy rainfall causes severe runoff that may increase the risk of pathogen diffusion, further worsening soil and water contamination³.”

Q4. I know that the authors took WASH into account and looked at risks by various patterns. Did they adjust for type of water supply and patterns by seasonality. Diarrhea and enteric infection burdens vary considerably by season, water supply interruption patterns in slums, periodic flooding etc. Were these investigated?

A4. Thank you for your questions. The DHS queries about source of drinking water for each household. The following types were categorized by the DHS as an *improved* source: piped into dwelling piped to yard/plot, public tap/standpipe, piped to neighbor, tube well or borehole, protected well, protected spring, rainwater, tanker truck, cart with small tank, and bottled water. The following types were categorized by the DHS as an *unimproved* source: unprotected well, unprotected spring, surface water (river/dam/lake/pond/stream/canal/irrigation channel), and other sources. We did not adjust for this variable in the main models considering the potential mediating effect of all WASH variables, but we assessed its effect modification (improved versus unimproved) for the investigated association (Supplementary Figure 11 in the Supplementary information). No significant difference in the drought-associated diarrhea risk between unimproved and improved water source was observed, except for 6- and 12-month mild drought, with a significantly stronger association observed for children with unimproved water source.

We did not test effect modification by season because in many regions, drought is thoroughly intertwined with season. However, we adjusted for seasonality of diarrhea by including a spline of the survey month in all models.

Water supply interruption patterns in slums were not investigated in our study due to data unavailability. In addition, this variable is likely to be on the pathway between drought and diarrhea occurrences and thus should not be controlled as a confounding factor. We now added a sentence in the Discussion: “Moreover, during drought, there also is the possibility of piped/tap water contamination due to increased turbidity from the loss of pressure at water treatment plants⁸, and epidemiological evidence has demonstrated the association between gastrointestinal

illness and water distribution system deficiencies during periods of interrupted supply⁹.” (Lines 218-221) As mentioned above, we added this as a limitation of this study: “Lastly, in addition to food safety, food insecurity, and migration discussed above, there are other factors by which diarrhea burdens could vary considerably, especially water supply interruption patterns in slums. Our analysis could not disentangle the interrelationship between drought and all these factors due to data unavailability.” (Lines 321-325).

Lastly, examining the associations between period flooding and diarrhea are beyond the scope of the current study. We plan to perform a detailed analysis of the relationship between flooding and diarrheal disease using DHS data in the future.

Reviewer 2

Q1. My main concern with the analysis is that the authors used the method of adjustment to look at mediation, which has many known weaknesses (see Richiardi et al, 2013, IJE for a summary), particularly because these variables might also be effect modifiers. The authors might want to consider an alternative method that considers interaction and mediation jointly in a more formal way. For example, the approach described by Vanderweele in doi: 10.1097/EDE.000000000000121 could be used here. If the authors feel that the method of adjustment is justified in this case, please provide further justification in the main text. Relatedly, I recommend that the authors move some of the discussion text about mediation vs. effect modification to the introduction section to introduce more fully the reason why these WASH variables might function in both ways, which I agree is possible. Adding some text to the introduction will help the reader to more easily follow the paper narrative.

A1. We really appreciate your valuable comments. Following your suggestions, we tried the method proposed by Vanderweele¹⁰ to assess the mediating effect by WASH variables. We looked into the implementation code of the referred mediation analysis and found that non-linear covariates and random effects are not allowed in the model, so we included mean temperature, rainfall, and survey year as linear terms, and the random effect variable for survey sites was removed. In addition, this method only allows the inclusion of a single mediator due to the calculation of interaction between the exposure and mediator. Therefore, we explored the

mediating effect of one WASH variable at a time. Please see the result for 6-month drought in
Table R1 below.

Table R1. Mediating effects of individual WASH variables (assessed in separate models) using the 4-way decomposition method allowing for interaction between 6-month drought and the WASH variable.

	Water source	Time to collect water	Toilet facility	Place to wash hands	Water availability	Soap availability	Water treatment
Mild drought							
Total excess RR (95% CI)	0.055 (0.049–0.061)	0.062 (0.056–0.067)	0.062 (0.056–0.068)	0.108 (0.103–0.112)	0.144 (0.139–0.149)	0.167 (0.164–0.172)	0.097 (0.090–0.101)
CDE (95% CI)	0.013 (-0.001–0.022)	0.047 (0.042–0.053)	0.008 (0.002–0.019)	0.057 (0.049–0.065)	0.061 (0.055–0.066)	0.067 (0.043–0.074)	0.086 (0.082–0.091)
INT_{ref} (95% CI)	0.040 (0.037–0.047)	0.008 (0.008–0.009)	0.049 (0.043–0.053)	0.022 (0.020–0.025)	0.056 (0.049–0.064)	0.086 (0.077–0.105)	0.011 (0.002–0.017)
INT_{med} (95% CI)	0.001 (0.001–0.001)	0.002 (0.001–0.002)	0.001 (0.001–0.002)	0.006 (0.005–0.006)	0.012 (0.011–0.014)	0.008 (0.007–0.010)	0.000 (0.000–0.000)
PIE (95% CI)	0.001 (0.001–0.001)	0.004 (0.004–0.005)	0.004 (0.004–0.004)	0.023 (0.022–0.024)	0.014 (0.012–0.015)	0.006 (0.005–0.006)	0.000 (0.000–0.001)
CDE proportion	23.3%	76.7%	13.0%	53.0%	42.7%	40.1%	88.4%
INT_{ref} proportion	72.6%	13.5%	78.8%	20.1%	38.9%	51.4%	10.9%
INT_{med} proportion	2.0%	2.6%	2.4%	5.2%	8.6%	5.0%	0.2%
PIE proportion	2.0%	7.2%	5.9%	21.8%	9.8%	3.6%	0.5%
Severe drought							
Total excess RR (95% CI)	0.116 (0.104–0.129)	0.131 (0.119–0.143)	0.131 (0.118–0.143)	0.236 (0.227–0.244)	0.340 (0.323–0.354)	0.383 (0.374–0.395)	0.204 (0.188–0.213)
CDE (95% CI)	0.026 (-0.002–0.044)	0.097 (0.085–0.108)	0.016 (0.003–0.039)	0.118 (0.101–0.134)	0.127 (0.114–0.136)	0.139 (0.087–0.154)	0.180 (0.170–0.190)
INT_{ref} (95% CI)	0.083 (0.078–0.098)	0.018 (0.016–0.020)	0.102 (0.092–0.111)	0.046 (0.043–0.052)	0.127 (0.109–0.145)	0.195 (0.175–0.239)	0.023 (0.005–0.038)
INT_{med} (95% CI)	0.005 (0.004–0.006)	0.007 (0.006–0.008)	0.006 (0.005–0.007)	0.024 (0.022–0.027)	0.058 (0.050–0.066)	0.037 (0.033–0.045)	0.001 (0.000–0.002)
PIE (95% CI)	0.002 (0.002–0.002)	0.010 (0.009–0.010)	0.007 (0.007–0.008)	0.048 (0.045–0.050)	0.029 (0.025–0.032)	0.012 (0.010–0.012)	0.001 (0.001–0.001)
CDE proportion	22.3%	73.6%	12.4%	49.8%	37.3%	36.3%	87.8%
INT_{ref} proportion	71.9%	13.5%	77.6%	19.6%	37.2%	51.0%	11.3%
INT_{med} proportion	3.9%	5.6%	4.6%	10.3%	17.0%	9.7%	0.5%
PIE proportion	1.9%	7.3%	5.5%	20.3%	8.6%	3.0%	0.5%

WASH: water, sanitation, and hygiene; RR: relative risk; CI: confidence interval; CDE: controlled direct effect; *INT_{ref}*: reference interaction; *INT_{med}*: mediated interaction; PIE: pure indirect effect/mediated main effect. These mediating effect estimates were calculated by a 4-way decomposition method¹⁰, in which the overall effect of drought can be decomposed into the proportion of the effect due to 1) the direct effect of drought (*CDE*); 2) only interaction between drought and WASH (*INT_{ref}*); 3) both mediation by WASH and the aforementioned interaction (*INT_{med}*); and 4) only mediation by WASH (*PIE*). *INT_{ref}* + *INT_{med}* means the overall effect attributable to interaction; *INT_{med}* + *PIE* means the overall effect mediated. Of note, this method does not allow non-linear covariates, random effects, or multiple mediators. Thus, mean temperature, rainfall, and survey year were included as linear terms, the random effect variable for survey sites was removed, and the mediating effect of each WASH variable was assessed in a separate model.

For all WASH variables, the proportion mediated ($INT_{med} + PIE$) ranged from 0.7-27.0% for mild drought and 1.0-30.6% for severe drought. However, because WASH variables may affect one another, these individual proportions mediated across WASH variables cannot be summed up for an overall mediated effect¹¹. In addition, the total effect varies across these 7 models (due to inclusion of a different WASH) so the proportions mediated ($INT_{med} + PIE$) cannot be directly summed.

Although the proportions are not directly comparable with our previous mediation analysis, in which all the WASH variables were considered together as a single mediator, we can arguably state that WASH practices mediated a low to moderate proportion of the association between drought and diarrhea, similar to our conclusion from the previous mediation analysis. That being said, since the model specification in this method differs considerably from our main models, we prefer to keep our original mediation analysis in the main method. However, we used this alternative method as a sensitivity analysis and described the limitation of our original method. We have made revisions in the Methods, Results, and Discussion:

Methods: “Fifth, we used an alternative method to examine the mediating effect of WASH variables, allowing for interaction between drought and these variables, in which the overall effect of drought can be decomposed into the proportion of the effect due to: 1) the direct effect of drought; 2) the interaction between drought and WASH alone; 3) both mediation by WASH and the aforementioned interaction; and 4) mediation by WASH alone¹⁰. However, this method does not allow non-linear covariates, random effects, or multiple mediators, so we included mean temperature, rainfall, and survey year as linear terms, removed the random effect variable of survey sites, and assessed the mediating effect of each WASH variable in a separate model. Because WASH variables potentially affect one another, their individual mediation proportions cannot be summed up for an overall mediated effect for all WASH variables.” (Lines 492-501)

Results: “Finally, similar to our main mediation analysis, our alternative mediation analysis (with a different set of model specifications) indicated that a low to moderate proportion of the overall effect of drought (0.8%–25.6% for mild drought and 0.9%–29.2% for severe drought) was mediated by WASH practices. However, these proportions represent the range for individual WASH practices (Supplementary Table 4), whereas the main mediation analysis assessed the proportion mediated by all WASH practices combined.” (Lines 186-191)

Discussion: “However, the traditional method we used for the mediation analysis is prone to bias with the presence of interaction between the exposure (i.e., drought) and mediator (i.e., WASH practices)¹². Recently developed methods employing a counterfactual framework either do not accommodate multiple mediators¹⁰ or are incapable of taking high-dimensional interactions (seven-way interaction in this study) into account¹³. Considering the study objective to explore the mediating nature of WASH, we ultimately adopted the traditional method, acknowledging the fundamental limitation in interpreting our mediation analysis results.” (Lines 233-239).

In addition, in the analysis shown in Table R1, we observed a high proportion of interactive effect from toilet facility. In our previous analysis, the model did not converge for 3 levels of toilet facility. Now we combine “unimproved facilities” and “open defecation” as a single group that we term “unimproved facilities”. The new models did not have the non-convergence problem, and we now report the results for its effect modification in Supplementary Figure 14.

As suggested, we also expanded the paragraph on WASH in the Introduction to give a clearer background on why we wanted to conduct both effect modification and mediation analysis by moving a sentence from the Discussion and by adding a sentence:

“Additionally, the shortage of clean water during drought may result in reduction in water availability for personal hygiene and sanitation, as adequate water for drinking needs to be prioritized, resulting in increased exposure to enteric pathogens. Therefore, WASH practices could play both an interacting and mediating role in the relationship between drought and diarrhea incidence.” (Lines 78-82 in Introduction).

Q2. Additionally, as a more minor comment, I wonder if the authors are adequately powered to look at the impact of 24-month droughts in tropical climates specifically. Could the authors provide some data (in the SI would be fine) about the number of drought months with data by climate zone? Figures S2-S4 are helpful, but because climate zone is not shown on the map I cannot be sure if the authors are adequately powered to test this interaction.

A2. We now have calculated summary statistics regarding drought-months experienced by children classified by climate zone. Please see the Table R2 below. We have also added this table to the Supplementary Information (Supplementary Table 1). In addition, note that the 95%

confidence interval for the relationship between 24-month drought and diarrhea risk in tropical climate zones are quite narrow, suggesting sufficient power (i.e., for mild drought, RR=1.06; 95% CI: 1.03–1.09; for severe drought, RR=1.01; 95% CI: 0.97–1.06).

Table R2. Descriptive statistics for the number of drought months experienced by included children^a during 1990–2019 by climate zone, drought severity, and drought timescale.

	Mean (SD)	Minimum	Median	Maximum	Interquartile range
Tropical zone					
Mild drought					
6-month	84.8 (9.7)	45	84	152	78–91
12-month	84.8 (13.2)	39	85	152	76–94
18-month	84.8 (14.2)	37	84	174	75–94
24-month	85.4 (17.1)	29	85	182	74–97
Severe drought					
6-month	37.3 (5.2)	11	38	57	34–41
12-month	37.8 (7.1)	4	38	64	34–42
18-month	37.5 (7.4)	4	38	68	33–42
24-month	37.2 (8.0)	6	37	67	32–42
Temperate zone					
Mild drought					
6-month	81.0 (8.7)	50	81	115	75–87
12-month	81.9 (13.4)	39	82	131	73–91
18-month	80.3 (13.0)	41	80	135	71–89
24-month	78.9 (15.2)	26	78	145	68–89
Severe drought					
6-month	37.0 (5.5)	18	37	57	34–41
12-month	36.1 (9.7)	2	37	63	31–43
18-month	35.8 (9.0)	2	37	63	31–42
24-month	36.1 (8.9)	5	37	67	31–42
Dry zone					
Mild drought					
6-month	83.2 (8.5)	53	84	112	78–89
12-month	85.6 (14.1)	33	86	132	76–96
18-month	83.9 (11.7)	37	84	128	76–92
24-month	82.5 (14.2)	26	83	130	73–92
Severe drought					
6-month	36.3 (6.5)	12	37	54	33–41
12-month	35.5 (9.6)	2	37	63	31–42
18-month	35.1 (8.8)	3	36	59	31–41
24-month	35.3 (8.8)	3	36	65	31–41

^a47,928, 26,818, and 33,143 missing values in the tropical, temperate, and dry zone respectively were not included.

Note that these statistics are for drought-months across all 1,379,566 children during the entire 30 years (1990-2019), with 47,928 (3.5%), 26,818 (1.9%), and 33,143 (2.4%) values missing in tropical, temperate, and dry zone, respectively. Thus, the statistics shown in the table are different from those in Figure 2, which shows the descriptive statistics across all 10km grids for the included countries in the raster of number of drought-months, and a grid may not be necessarily matched to a child.

Based on this table, children experienced an average of 85.4 (SD: 17.1) months of mild drought and 37.2 (SD: 8.0) months of severe droughts at the 24-month timescale in tropical climates. Similar numbers of drought-months can be observed in other time scales and other climate zones, showing that climate zone was not linked to the frequency or severity of drought (i.e., children living in dry zones did not experience drought more frequently).

We also calculated the summary statistics regarding drought-months experienced by children classified by climate zone using the dataset we used to perform the effect modification by climate zone ($N=709,412$, no missing values). Please see Table R3 below. We don't find substantial discrepancies between the results from these two datasets.

Table R3. Descriptive statistics for the number of drought months experienced by included children ($N=709,412$) during 1990–2019 by climate zone, drought severity, and drought timescale.

	Mean (SD)	Minimum	Median	Maximum	Interquartile range
Tropical zone					
Mild drought					
6-month	83.7 (9.8)	45	83	152	77–90
12-month	84.0 (13.4)	39	84	152	75–93
18-month	84.0 (14.3)	37	83	174	74–93
24-month	84.3 (17.4)	30	84	182	72–96
Severe drought					

6-month	36.7 (5.8)	11	37	57	33–41
12-month	36.9 (7.6)	4	38	61	33–42
18-month	36.8 (7.8)	4	37	68	32–42
24-month	36.6 (8.5)	6	37	67	31–42
Temperate zone					
Mild drought					
6-month	79.9 (8.4)	50	80	115	74–85
12-month	81.3 (13.3)	39	81	131	73–90
18-month	79.4 (13.5)	41	79	127	70–88
24-month	78.1 (15.2)	26	77	134	67–89
Severe drought					
6-month	36.2 (5.7)	18	37	56	33–40
12-month	34.9 (10.4)	2	36	63	29–42
18-month	34.7 (9.7)	2	36	61	29–41
24-month	35.1 (9.3)	5	36	67	30–41
Dry zone					
Mild drought					
6-month	82.6 (8.5)	53	83	112	77–89
12-month	86.0 (14.4)	33	87	132	77–96
18-month	83.5 (11.8)	37	84	128	75–92
24-month	82.0 (14.5)	26	82	130	72–92
Severe drought					
6-month	35.5 (7.0)	12	37	54	31–41
12-month	34.0 (10.8)	2	36	63	28–41
18-month	33.6 (9.9)	3	36	59	28–40
24-month	33.7 (9.5)	3	35	65	28–40

We have included the following sentence in the Result section:

“We calculated the number of drought-months each included child experienced and found the number of months with severe drought to be substantially lower than the number with mild drought (Supplementary Table 1). However, we did not observe noticeable differences in drought-months experienced by children across timescales or climate zones.” (Lines 112-116).

In addition to the responses above, we noticed a calculation error regarding the number of drought-months, and we have revised Figure 2 and Supplementary Figures 1–4 accordingly.

References

1. Hermans K, McLeman R. Climate change, drought, land degradation and migration: exploring the linkages. *Curr Opin Env Sust* **50**, 236-244 (2021).
2. Sam AS, Abbas A, Padmaja SS, Kaechele H, Kumar R, Muller K. Linking food security with household's adaptive capacity and drought risk: implications for sustainable rural development. *Soc Indic Res* **142**, 363-385 (2019).
3. Tirado MC, Clarke R, Jaykus LA, McQuatters-Gollop A, Franke JM. Climate change and food safety: a review. *Food Res Int* **43**, 1745-1765 (2010).
4. Cooper MW, *et al.* Mapping the effects of drought on child stunting. *Proc Natl Acad Sci U S A* **116**, 17219-17224 (2019).
5. Duchenne-Moutien RA, Neetoo H. Climate change and emerging food safety issues: a review. *J Food Protect* **84**, 1884-1897 (2021).
6. Milićević D, *et al.* Climate change: impact on mycotoxins incidence and food safety. *Theor Pract Meat Process* **4**, 9-16 (2019).
7. World Health Organization. Mycotoxins. (2018).
8. Yusa A, *et al.* Climate change, drought and human health in Canada. *Int J Environ Res Public Health* **12**, 8359-8412 (2015).
9. Ercumen A, Gruber JS, Colford JM, Jr. Water distribution system deficiencies and gastrointestinal illness: a systematic review and meta-analysis. *Environ Health Perspect* **122**, 651-660 (2014).
10. VanderWeele TJ. A unification of mediation and interaction: a 4-way decomposition. *Epidemiology* **25**, 749-761 (2014).
11. Lash TL, VanderWeele TJ, Haneuse S, Rothman KJ. Mediation Analysis. In: *Modern Epidemiology*. 4th edn (2020).
12. Richiardi L, Bellocco R, Zugna D. Mediation analysis in epidemiology: methods, interpretation and bias. *Int J Epidemiol* **42**, 1511-1519 (2013).
13. Bellavia A, Valeri L. Decomposition of the Total Effect in the Presence of Multiple Mediators and Interactions. *Am J Epidemiol* **187**, 1311-1318 (2018).

Reviewers' Comments:

Reviewer #1:

Remarks to the Author:

I believe the authors have responded fairly and robustly to all the queries and comments. I have no further queries and believe that this scholarship will add valuable information to available data in this field.

Reviewer #3:

Remarks to the Author:

I thank the authors for their thorough response to my comments. The authors have adequately addressed my comments and I have no further concerns. It is nice to see the improved mediation analysis in SI and it is encouraging that the results generally agree with the models from the main text.

Response to the editor and reviewers

Reviewer #1

I believe the authors have responded fairly and robustly to all the queries and comments. I have no further queries and believe that this scholarship will add valuable information to available data in this field.

We thank Reviewer 1 for their considerate and detailed comments and suggestions for this study, which significantly improved the current manuscript for publication.

Reviewer #3

I thank the authors for their thorough response to my comments. The authors have adequately addressed my comments and I have no further concerns. It is nice to see the improved mediation analysis in SI and it is encouraging that the results generally agree with the models from the main text.

We sincerely appreciate the thoughtful and constructive suggestions and comments from Reviewer 3, particularly for the mediation analysis, which undoubtedly add further strength to this study.

We also would like to take the opportunity to thank the editors for all the editorial advices and comments for the smooth publication on Nature Communications.